# The Effect of the Promotion of Vegetables by a Social Influencer on Adolescents’ Subsequent Vegetable Intake: A Pilot Study

**DOI:** 10.3390/ijerph17072243

**Published:** 2020-03-26

**Authors:** Frans Folkvord, Manouk de Bruijne

**Affiliations:** 1Open Evidence Research, Barcelona, 08018 Barcelona, Spain; 2Tilburg School of Humanities and Digital Sciences, Communication and Cognition, 5037 AB Tilburg, The Netherlands; 3Independent Researcher, De Boelelaan 1101, 1081 HV Amsterdam, The Netherlands

**Keywords:** food cues, food marketing, healthy food, adolescents

## Abstract

Marketers have found new ways of reaching adolescents on social platforms. Previous studies have shown that advertising effectively increases the intake of unhealthy foods while not so much is known about the promotion of healthier foods. Therefore, the main aim of the present experimental pilot study was to examine if promoting red peppers by a popular social influencer on social media (Instagram) increased subsequent actual vegetable intake among adolescents. We used a randomized between-subject design with 132 adolescents (age: 13–16 y). Adolescents were exposed to an Instagram post by a highly popular social influencer with vegetables (n = 44) or energy-dense snacks (n = 44) or were in the control condition (n = 44). The main outcome was vegetable intake. Results showed no effect of the popular social influencer promoting vegetables on the intake of vegetables. No moderation effects were found for parasocial interaction and persuasion knowledge. Bayesian results were consistent with the results and supported evidence against the effect of the experimental condition. Worldwide, youth do not consume the recommended amount of fruit and vegetables, making it important to examine if mere exposure or different forms of food promotion techniques for healthier foods are effective in increasing the intake of these foods.

## 1. Introduction

Eating sufficient nutritious foods is essential for a healthy growth and development. More specifically, vegetables protect against many illnesses and chronic diseases and improves psychological well-being [1,2]. Next to that, studies have shown that eating behaviors established during youth tracks into adulthood and contributes to long-term health and disease [2,3]. Nonetheless, studies show that dietary intake patterns of adolescents in the Netherlands do not meet (inter)national dietary standards [3]. In particular, Dutch adolescents do not eat enough fruits and vegetables [3]. Moreover, evidence has shown that traditional interventions have limited effect, such as educational programs or parental dietary programs, and new innovative promotional methods and techniques need to be tested to make fruit and vegetables more appealing among adolescents [4].

Next to that, the popularity of social media among adolescents keeps rising and following blogs and vlogs (video blogs) has become hugely popular among millions of young people worldwide, thereby significantly influencing their daily behavior [5]. Due to the increasing popularity of online platforms such as YouTube, Facebook, and Instagram, food advertisers have found new and unique ways of targeting children and adolescents on these platforms, affecting their consumption behavior and subsequent dietary intake [6,7,8,9,10]. Currently, a common food marketing technique to advertise for energy-dense foods is influencer marketing.

The exponential growth of social media has given rise to so-called microcelebrities, such as bloggers or vloggers, who have gained fame on social media through self-branding and mere exposure [11]. These new types of celebrities, labeled as social influencers, use social media to engage in strategic self-presentation to attract attention and followers among large numbers of youth [12]. Through the use of social media, social influencers are able to reach thousands or even millions of potential customers and are thus highly influential, in particular among children and adolescents that identify with the influencers, through parasocial interaction [6]. Influencer marketing involves the action of promoting and selling products or services through social influencers who have the capacity to have an effect on the character and perception of a brand. In short, influencer marketing builds brand alliances between influencers and their followers to promote their products or services [13,14,15].

Based on general cognitive and developmental psychology [16,17], one’s (digital) environment is a strong influencer of someone’s consumption behavior. More specifically, according to Social Learning Theory [16], modeling, reinforcement, and social interactions are mechanisms that explain how adolescent’s acquire cognitions and behaviors from their social agents. Translating these insights to the current communication forms of youth, scholars have shown that these online influencers have a stronger impact on consumer socialization as “social influencers” because they are considered as peers and laymen, which reduces the activation of persuasion knowledge, whereby recipients perceive a “parasocial interaction” with the online celebrities and thus perceive them as more credible than traditional celebrity endorsers [17].

In addition, the Processing of Commercial Media Content Model (PCMC-model [3]) has provided more insight in the processing of influencer marketing messages by young people. The PCMC-model has integrated multiple theoretical frameworks, adopting a developmental perspective on adult persuasion models [18,19] and theories of adolescents’ consumer development and socialization [20,21,22]. In short, the PCMC-model theorizes how communication messages can predict persuasion processing based on a limited capacity information approach [7]. Following the PCMC-model, for example, factors like proximity, prominence, interactivity, and the level of integration of the messages in the content reduce the level of persuasion knowledge activation and increase the level of parasocial interaction.

More specifically, adolescents’ motivation and ability to critically process the marketing message of the social influencer will be relatively low in comparison to more traditional forms of advertising because the advertising cues in social influencer messages are highly integrated in the entertaining content. Therefore, the cognitive resources of adolescents’ automatically will be used to process the entertaining aspect of the message, while additional sources are needed to activate skepticism towards the advertising element of the message. Considering that most adolescents will not be motivated to do so, skepticism will not be activated among the recipients [7].

According to the Differential Susceptibility to Media Effects model (DSMM [23]), the level of cognitive elaboration of the cues in advertising leads to cognitive, emotional, and behavioral changes towards the advertised product and brand [9]. The basic underlying mechanism of the DSMM is that it shows why some adolescents are more susceptible to media effects than others; in this study, that adolescents have a lower level of persuasion knowledge and higher level of parasocial interaction; and how and why media influences these individuals [23].

Previous studies have shown that multiple advertising techniques are effectively increasing the intake of unhealthy foods, also by social influencers [10], while not so much is known for the effects of the promotion of healthier foods [6,9,24,25,26]. A great number of studies have shown that the intake of fruits and vegetables among adolescents lies decidedly below recommended guidelines [27,28]. Because dietary patterns of children and adolescents continue into adulthood, targeting adolescents’ fruit and vegetable intake improves healthy lifestyles across the life span and has a strong effect on noncommunicable diseases and mental well-being [29,30]. Therefore, motivating and inducing adolescents to eat more fruits and vegetables, for example, through effective food promotion techniques that induce the intake of fruits and vegetables [25], will regain and maintain healthy weight statuses and improves other health indicators (e.g., inflammation levels, blood lipids, blood pressure, insulin sensitivity, and mental well-being [30,31,32,33]). In addition, the intake of fruits and vegetables is negatively related to overweight and obesity [34], psychological dysfunctioning [35], risk of cardiovascular disease [36], coronary heart disease and hypertension [37], and multiple forms of cancer [38].

### 1.1. Promoting Healthy Foods on Social Media

One theoretical framework that aims to predict and explain the effects of the promotion of healthy foods is the Promotion of Healthy Food Model [5]. This theoretical framework states that, by increasing attention toward and thereby reinforcing the value (e.g., liking and wanting [39]) of fruits and vegetables through food promotion, adolescents will consume more fruits and vegetables and a reciprocal relation with eating behavior occurs, which in time leads to a normalization of intake of fruits and vegetables. Importantly, the Promotion of Healthy Food Model states that individual and societal factors determine susceptibility and effectiveness of food promotion and should be examined to get a better understanding of the effectiveness of food promotion techniques of healthier foods [5].

Individual susceptibility to advertising messages effects may in particular be considered when it comes to influencer marketing by social influencers, as these social influencers specifically design their messages for existing community of engaged and young followers, thereby reducing their persuasion knowledge activation and increasing the parasocial interaction, leading to a more positive attitude towards the advertised product and increasing purchase intentions [40,41,42]. In addition, promotion activities by peer-aged influencers are perceived as more credible and authentic because adolescents believe they will receive an honest advice or opinions about certain brands or products. Therefore, the probability that adolescents will activate skepticism towards the social influencer is less than in more traditional forms of advertising, such as advertising on television or in magazines [5,7].

For advertising on Instagram, where social influencers often involve viewers in their personal life, for example, by sharing intimate personal stories, this identification process is often reinforced and can lead to conformity among the viewers, through parasocial interaction [43]. Similarly, the home situation that is seen in the advertising can further increase identification with the social influencer and increase the likelihood that the product that is endorsed is viewed more positively, leading to an increased consumption of the advertised brand and product [44].

### 1.2. Current Study

The main aim of the present experimental pilot study was to test if promoting red peppers by a popular social influencer on social media (Instagram) increased subsequent actual vegetable intake among adolescents compared to adolescents who were exposed to the promotion of energy-dense snack by a popular social influencer on social media or to the promotion of nonfood products by a popular social influencer on social media. The expectation was that adolescents who were exposed to the promotion of red peppers were more likely to choose vegetables than the adolescents who were exposed to the promotion of energy-dense snacks or nonfood products. Furthermore, it was expected that the level of parasocial interaction and persuasion knowledge moderates this effect.

## 2. Materials and Methods

### 2.1. Experimental Design and Stimulus Material

In the current experimental pilot study, we used a between-subjects design with three conditions (type of Instagram post: vegetables vs. energy-dense snacks vs. nonfood products). The manipulated posts were all derived from a highly popular Dutch social influencer, named Kalvijn, and were identical regarding the amount of likes (N = 12,543), reactions (N = 65), and the explanatory text, except for the products presented. The Instagram post was integrated in a paper questionnaire that was distributed to the adolescents. Adolescents were randomly assigned to one of the three conditions. The vegetable condition was an Instagram post by Kalvijn of red pepper. The energy-dense condition was an Instagram post by Kalvijn of finger foods. The nonfood condition was an Instagram post by Kalvijn of sunglasses. One of the vegetables provided for consumption was also used in the Instagram post (red pepper). The dependent variable in this study was vegetable intake. Adolescents could choose of three vegetables, namely small red peppers, cucumbers, and cherry tomatoes. These snack foods are very popular among adolescents in the Netherlands. Adolescents (N = 132) were told that they could choose freely from the vegetables and could take as many as they wanted. We used more vegetables snacks than were shown in the Instagram post because previous studies have shown that food advertising leads to spillover effects as well [9,25,26]. In this context, spillover effects mean that an advertisement for a certain brand or product leads not only to an increase of the advertised brand or product but also to an increased intake of different brands or products that fall in the same food-category.

### 2.2. Participants

All adolescents were in the first or second grade of secondary school. Adolescents participated collectively within their own group, during school hours in the presence of their teachers and the research assistant. Before the assessment, we emphasized to both parents and the adolescents that all of the data for this study would remain confidential and that adolescents could cease participation at any time. The experiment took approximately 15–20 min. The study protocol was reviewed and approved by the Ethical Board of the Communication Science Department at the Vrije Universiteit Amsterdam. Data for this study were collected in May 2018. The data that supports the findings of this study are available from the corresponding author upon reasonable request.

### 2.3. Procedures

After obtaining written consent from both schools to participate in the current study, we sent all parents a letter with detailed information regarding the study and we asked them to actively consent for participation of their child(ren). Before the assessment, we emphasized to both parents and the adolescents that all of the data for this study would remain confidential and that they could cease participation at any time. The cover story was that we were interested in the media use of youth. Adolescents in both grades filled out the questionnaires in a classroom individually with the research assistant present to clarify potential difficulties. Adolescents could not see other adolescents’ answers because the set-up of the classroom was conducted so that every adolescent was seated at a separate table, with enough room between tables.

The questionnaire consisted of two different parts. In the first part, adolescents answered questions regarding sociodemographical questions, such as age and gender, followed by a question about their hunger level (masked by questions about their level of thirst and arousal). Next, their general usage of Instagram, familiarity with social influencers, and whether they follow any of these social influencers were assessed. Subsequently, adolescents were exposed to the Instagram post at the end of this part of the questionnaire. Participants were told that they should pay attention to the post because subsequently questions regarding the post would follow, which took 1–2 min on average. After finishing the first part of the questionnaire, they handed in the questionnaire to the research assistance, where they could take a snack (vegetables snacks: little red peppers, cucumbers, and tomatoes). The research assistant explained that they could take as much snacks as they liked, ad libitum. Then, the research assistant provided the second part of the questionnaire to the participants. When the child returned to their seat, the research assistant coded (encrypted) which and how many vegetables were consumed by the participant without notifying the adolescents that this was coded. The second part of the questionnaire consisted of questions related to parasocial interaction with the social influencer, persuasion knowledge, general vegetable intake, favourite vegetable, and the manipulation check.

### 2.4. Measures

Vegetable consumption: The dependent variable was vegetable consumption. Actual consumption was assessed by counting the type and amount of vegetables that were consumed as a snack.

Parasocial interaction. Parasocial interaction was assessed by thirteen statements that were adapted for the current study, derived from the Audience-Persona Interaction (API) Scale [45], also used for assesses parasocial interaction with characters on television [46]. Statements were: “Kalvijn looks like me”, “I have the same qualities as Kalvijn”, “I have the same interests as Kalvijn”, “I have the same problems as Kalvijn”, “I find it important to see what Kalvijn is doing in his life”, “I like to hear Kalvijn’s voice”, “I react the same to Kalvijn as to my friends”, “I react the same to Kalvijn as to my family”, “My friends look like Kalvijn”, “Kalvijn could have been one of my friends”, “I agree often with Kalvijn”, “I can identify myself with Kalvijn”, and “I could be friends with Kalvijn”, that were scored on a five-points scale with the anchors going from “I totally disagree” to “I completely agree”. The Cronbach’s alpha of this scale was 0.96. 

Persuasion knowledge. Persuasion knowledge was assessed with five items on a seven-point scale, derived from Carter et al. [47]. Items were “Kalvijn wants his followers to like a product”, “Kalvijn wants his followers to use a product”, “Kalvijn wants to influence his followers”, “Kalvijn wants to keep his followers up-to-date”, and Kalvijn wants to entertain his followers”. The Cronbach’s alpha of this scale was 0.88.

### 2.5. Statistical Analysis

First, data was tested for normality and homogeneity of variance (Levene’s test). Next, before testing our hypotheses, we conducted randomization checks with a multivariate analysis of variance (MANOVA) for gender, age, hunger, parasocial interaction, and persuasion knowledge. Table 1 presents the means and standard deviations for all variables separately for each condition. Then, we computed residual scores; tested them for Mahal’s distance, Cook’s distance, and leverage scores; and found no indication to assume outlying scores. Furthermore, to check for potential covariates, Pearson’s correlations between the model variables were assessed. No factors correlated significantly with the amount of vegetables consumed (*p* > 0.05), so no covariates were included in the analyses.

In addition, univariate analysis of variance (ANOVA) tested the effect of the type of Instagram post on total vegetable intake and multivariate analysis of variance (MANOVA) tested the effect of the type of Instagram post on intake of the three types of vegetables. To further test for the (non)existence of the main effects of the experimental condition, Bayesian ANOVA was performed with the statistical program JASP [48]. Evidence for each model in this analysis is evaluated against the null model. Following conventional interpretation, a value of BF10 above 3 is interpreted as substantial support for the alternative hypothesis and a value of BF10 less than 0.33 was substantial support for the null hypothesis. BF10 values between 0.33 and 3 suggest the data are insensitive (e.g., Reference [49]). The adjusted one-sided *p* value that was considered significant was 0.05.

## 3. Results

In the current study, 132 adolescents participated (age: 13–16 y; M = 14.1, SD = 0.96; 46.2% was girl). No significant differences were found between the experimental conditions for gender, age, hunger, parasocial interaction, and persuasion knowledge (*p* > 0.05), so randomization seemed successful. The manipulation check consisted of one question assessing which product the child had seen in the Instagram post. The results showed that 89% answered correctly on this question. More specifically, 91% in the vegetable condition, 93% in the energy-dense condition, and 82% in the nonfood condition answered correctly. A Chi-square analysis showed that the results were not significantly different between the conditions. Distribution of the data was normal and the Levene’s test showed no significant differences in homogeneity of variance.

The results from the ANOVA showed no significant main effect of type of Instagram post on vegetable intake (*p* > 0.05, eta squared = 0.023). Bayesian ANOVA was consistent with this result and supported evidence against the effect of the Instagram post (BF10 = 0.095). In addition, the MANOVA showed no significant effect of type of Instagram post on the three individual vegetable consumption (*p* > 0.05).

In addition, no significant interaction effect of type of Instagram post and parasocial interaction on vegetable intake (*p* > 0.05, eta squared = 0.193) was observed. Next, no significant interaction effect of type of Instagram post and persuasion knowledge on vegetable intake (*p* > 0.05, eta squared = 0.200) was observed. Again, Bayesian ANOVA was consistent with these results and supported evidence against the effect of the Instagram post and the interactions, respectively (BF10 = 0.014) and (BF10 = 0.031). These results do not confirm the hypotheses.

## 4. Discussion

The main aim of the current experimental pilot study was to examine if promoting red peppers by a popular social influencer on social media (Instagram) increased subsequent actual vegetable intake among adolescents compared to adolescents who were exposed to the promotion of energy-dense snack or nonfood products. The expectation was that adolescents who were exposed to the promotion of vegetables were more likely to choose vegetables than the adolescents who were exposed to the promotion of energy-dense snacks or nonfood products. Furthermore, it was expected that the level of parasocial interaction and persuasion knowledge would moderate this effect.

The results showed no effect of the popular social influencer promoting red peppers on the choice or intake of vegetables. In addition, no moderation effects were found for parasocial interaction and persuasion knowledge. Bayesian results were consistent with the results and supported evidence against the effect of the experimental condition. A great amount of research has shown that food marketing affects actual consumption, but studies examining the effects of the promotion of healthy foods is limited [6,24]. Taking into account that youth worldwide do not consume the recommended amount of fruit and vegetables, it is important to examine if mere exposure or different forms of food marketing are effective in increasing the intake of healthier foods. Until now, it is unclear if promotion techniques for healthy foods have an effect on the intake of healthier foods [24,26].

In order to improve adolescents’ eating behavior and to reduce the number of adolescents that become overweight or obese, it is necessary to test effective methods to make healthy foods more appealing and to subsequently increase the intake of these food products, preferably substituting energy-dense foods [50]. Existing evidence shows that food advertising affects eating behavior among adolescents, but most research has been focusing on the effects of unhealthy food advertising on adolescents’ eating behavior [5]. The Promotion of Healthy Food Model can be used as a theoretical framework that links up existing empirical evidence on how food promotion techniques influence eating behavior and provides future research questions as well as intervention opportunities. The comprehensive theoretical model [5] can be used to create new strategies and programs that can be part of a greater food transformation as has been proposed by a large group of scientific researchers to improve sustainable and healthy dietary behavior [51]. Nonetheless, it should be further examined if, when, for whom, and how food promotion techniques for healthier foods are effective [5].

One of the strengths of the current pilot study is that it is one of the first investigating the effect of a social influencer on the intake of vegetables. A large number of studies have been conducted examining the effects of unhealthy food marketing on the attitudes, intentions, preferences, and actual intake of adolescents (see also References [5,7]), but only a few studies have been conducted examining the effect of healthier food promotions (see for an overview References [5,7]). Second, we examined the effect on actual intake of vegetables. Considering the great benefits of vegetable intake among this target group, it is of great importance that not only cognitive and emotional responses to food advertising are examined but also, or maybe even of more importance, actual intake should be examined. Third, we used existing Instagram posts posted by a very popular social influencer among the target group in the Netherlands, thereby guaranteeing ecological validity. One limitation of the current study is that we only tested one Instagram post. Normally, children watch multiple posts during their activity on social media. Considering that we used actual posts, it was very difficult to obtain multiple posts with the same content, thereby taking into account the importance of controlling for the similarity of the content of the stimulus material. A second limitation is that we have used a paper format for the Instagram posts, while in reality, adolescents watch Instagram posts online with an audiovisual format. Therefore, recommendations for future research is that adolescents should be exposed to multiple Instagram posts in an audiovisual format, thereby taking into account the similarity of the content of the stimulus materials for the different conditions. Third, the Instagram post showed only one vegetable: red pepper. Potentially, some children did not like red peppers, which could have affected the results. Therefore, future research could use more different forms of vegetables in order to establish difference effects between the exposure of different vegetables.

### Recommendations for Future Research

First, one important line of research could for example examine the mechanism underlying the effects of food promotion techniques on actual intake. More specifically, it is important to investigate whether the promotion of fruits and vegetables influences an increased attention for these foods and, if this reinforces the perceived value and actual craving, by studying physiological and psychological responses when children are exposed to these promotional techniques [9]. For example, Ferriday and Brunstrom [52] showed that adults with obesity show increased salivary responses and craving after being exposed to unhealthy food cues compared to adults with normal weight. Whether the same effects can be found for cues of healthier foods is unknown [5]. In addition, it would be interesting to examine if brain areas relating to the reward system are activated when children are exposed to the promotion of fruits and vegetables [53,54,55] and if this consequently increases the actual intake of these foods because of the intensified rewarding value.

Previous studies have found an effect of the promotion of healthy foods on intake among adolescents [56,57,58,59]. In addition, several studies have found a neurological effect of food advertising for unhealthy foods in adolescents [53,54], but this has not yet been tested for promotion techniques for healthy foods. For example, Gearhardt et al. [54] have shown that adolescents exhibited greater activation in regions implicated in reward areas during exposure to unhealthy food promotions, but this effect has not yet been tested using healthy food promotions and could be very informative in this line of research. In addition, according to the PCMC-model, specific message characteristics affect persuasion processing through different levels of cognitive elaboration that depend on the recipients’ level of attention and awareness of the message and on their motivation and ability to process the message effectively. More specifically, according to the PCMC-model, adolescents’ motivation and ability to critically process the marketing message of social media influences should be lower compared to other forms of advertising, such as television or billboards, because the advertising cues in social influencer messages are highly embedded in the entertaining content. Therefore, it is expected that the effect of social media marketing will be stronger than regular food advertising [9], but this has not been examined thoroughly before. Nonetheless, because of the experimental set-up of the current study, using questionnaires and a research assistance present in the classroom, it could have influenced the motivation to critically process the persuasion messages of the food promotion and therefore reduced the effect of the food promotion. Future studies should examine the effects of an audiovisual Instagram post promoting vegetables on the intake of an individual child or adolescent without other adolescents around that could potentially influence their consumption behavior.

Second, future research should examine whether and how the accumulation of food promotion techniques for healthy foods influences the classical and operant conditioning of food cues and subsequent intake of healthier foods among adolescents. Theoretical models that explain the effects of food marketing on intake consider the effects of food promotion techniques on food intake (via cue-reactivity) as a process of classical and operant conditioning [5,10], but it is unclear yet what the long-term effects of food promotion techniques of healthy food are and whether they lead to improved health indicators. In the end, social media marketing for healthy foods will not be the golden bullet that will change adolescents’ dietary intake instantly, but it could be one important part of the puzzle.

Third, although multiple individual (e.g., BMI, impulsivity, attentional bias, neophobia, and food fussiness) and contextual factors (e.g., socioeconomic status and parental feeding techniques) have been investigated that affect the influence of food promotion of unhealthy foods on eating behavior [5,10], it remains unclear if these factors moderate the effect of promotion of healthy foods. Establishing these and other individual and contextual dispositional factors, such as general vegetable intake or adolescents’ responses to the Food Frequency Questionnaire for children is vital for two reasons: first, for the development of new theoretical models that explain future findings and, second, in understanding the variability in the processing of the promotion of fruits and vegetables and the attention for and reinforcing value of these foods as well as subsequent intake [5].

## 5. Conclusions

To conclude, in the food marketing domain, some experts counsel against using the same promotional techniques that are used by food companies that market unhealthy foods because of their potential to undermine intrinsic motivation to obtain and consume healthier foods. As a result, using such techniques may backfire for some adolescents as well [5,10]. Therefore, political and societal discussions should be held in order to create more public awareness and to generate support for implementing contextual modifications. Political and societal debate on these and related issues is imperative [5]. The strategies applied to date by governments, schools, parents, and other stakeholders concerning the nutrition of adolescents knowingly or unknowingly affect adolescents’ dietary behavior in ways that are positive (e.g., increased dietary variety and intake of healthier foods and decreased pickiness and neophobia), negative (e.g., decreased intake of healthier foods and increased levels of neophobia), or simply have no effect on children’s eating behavior at all. Given the great focus on public health as well as parents´ controlling approach when it concerns their adolescents’ food intake, it is vital to rethink and reflect upon different and effective approaches to changing adolescents eating behavior in order to improve their dietary intake on the long-term.

## Figures and Tables

**Table 1 ijerph-17-02243-t001:** Variables measured by the condition ^a.^; (*N* = 132).

				Vegetable			Energy-dense		
				Nonfood					
	*n*	*M*	*SD*	*n*	*M*	*SD*	*n*	*M*	*SD*
Gender (girl)	44	45%	±0.50	44	48%	±0.51	44	45%	±0.50
Age (years)	44	14.14	±0.96	44	14.27	±0.92	44	14.02	±1.00
Hunger (VAS^a^)	44	5.57	±2.98	44	6.09	±3.60	44	6.23	±3.62
Parasocial interaction	44	1.84	±0.97	44	2.11	±1.14	44	1.74	±0.89
Persuasion knowledge	44	4.12	±1.87	44	3.73	±1.81	44	3.47	±1.69
Red peppers (number)	44	0.30	±0.59	44	0.30	±0.51	44	0.30	±0.77
Cucumbers (number)	44	0.55	±1.45	44	0.41	±0.66	44	0.39	±0.54
Cherry tomatoes (number)	44	0.11	±0.39	44	0.14	±0.67	44	0.02	±0.15
Vegetables total (number)	44	0.95	±1.56	44	0.84	±1.22	44	0.70	±1.00

^a^ Mean ± SD (all such values). VAS, visual analogue scale.

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
