# Peer review of "The Effect of the Promotion of Vegetables by a Social Influencer on Adolescents’ Subsequent Vegetable Intake: A Pilot Study"

_ijerph, 2020, doi:10.3390/ijerph17072243_

Round 1

Reviewer 1 Report

Critique

The investigators are to be commended for their efforts. As they noted, it is imperative to find a better way to promote healthy diets in children.

This is a clever study designed to evaluate if celebrities on social media (i.e., social influencers) can be leveraged to promote healthy diets. The investigators also present a clear and succinct summary of developmental psychology and marketing that supports their approach.

Unfortunately, the results showed no effect of the social media posts on the consumption of vegetables by the children in the study. Hopefully, this is only the beginning of this line of research for the investigators. Other studies need to be conducted to show whether social media can positively impact diet in children.

Major Comments

There does not appear to be a sample size analysis or power analysis. Perhaps the investigators considered this a pilot study to demonstrate that this approach is feasible before seeking outside funding to support a larger study. If so, this should be clearly reported as a pilot study. The authors do state that this study, “is one of the first investigating the effect of a social influencer on the intake of vegetables.” This would suggest that they do consider this study to be a pilot study. Regardless, it is possible that a larger sample size would be needed to show a significant difference in vegetable consumption between the groups.

The investigators chose an unusual vegetable for the study. Red peppers, or any peppers, are not the typical vegetable most children would consume. Of course, this probably differs by region, culture, family background, etc. However, the investigators had to choose from the available social media posts. Nevertheless, the findings of the study might have been very different if a different vegetable had been presented by the social influencer or if different vegetables were offered for the snack during the research study. Also, many children eat their vegetables with a dip or other food substance like peanut butter. Only vegetables were offered during the research study. It is not clear if a beverage was available to the children and this might also have impacted vegetable consumption. Furthermore, a fruit option, like an apple or banana, might have had more impact on diet than a vegetable.

The investigators also note that they, “only tested one Instagram post.” It seems unlikely that one post showing a social influencer eating a vegetable once is enough to outweigh the impact of years of inadequate vegetable consumption and advertising of unhealthy food options. Clearly, there are many other possibilities to consider with regard to promotion of healthy diet in children.

Minor Comments

None

Author Response

Critique

C1: The investigators are to be commended for their efforts. As they noted, it is imperative to find a better way to promote healthy diets in children. This is a clever study designed to evaluate if celebrities on social media (i.e., social influencers) can be leveraged to promote healthy diets. The investigators also present a clear and succinct summary of developmental psychology and marketing that supports their approach. Unfortunately, the results showed no effect of the social media posts on the consumption of vegetables by the children in the study. Hopefully, this is only the beginning of this line of research for the investigators. Other studies need to be conducted to show whether social media can positively impact diet in children.

A1: We thank the reviewer for reviewing the article and for the positive words. And of course, we will continue this line of research because of its great importance.

Major Comments

C2: There does not appear to be a sample size analysis or power analysis. Perhaps the investigators considered this a pilot study to demonstrate that this approach is feasible before seeking outside funding to support a larger study. If so, this should be clearly reported as a pilot study. The authors do state that this study, “is one of the first investigating the effect of a social influencer on the intake of vegetables.” This would suggest that they do consider this study to be a pilot study. Regardless, it is possible that a larger sample size would be needed to show a significant difference in vegetable consumption between the groups.

A2: We agree with the reviewer and have changed the title and the text accordingly.

C3: The investigators chose an unusual vegetable for the study. Red peppers, or any peppers, are not the typical vegetable most children would consume. Of course, this probably differs by region, culture, family background, etc. However, the investigators had to choose from the available social media posts. Nevertheless, the findings of the study might have been very different if a different vegetable had been presented by the social influencer or if different vegetables were offered for the snack during the research study. Also, many children eat their vegetables with a dip or other food substance like peanut butter. Only vegetables were offered during the research study. It is not clear if a beverage was available to the children and this might also have impacted vegetable consumption. Furthermore, a fruit option, like an apple or banana, might have had more impact on diet than a vegetable.

A3: Red peppers is one of the most popular vegetables in the Netherlands, next to carrots, cucumbers and tomatoes. Maybe the reviewer is mistaken by spicy red peppers, which were not included in the current study. In the Netherlands, these foods are mostly consumed without any dip or other foods. More specific, the snack vegetables we presented are specifically designed to eat as a snack for children. To be more clear we have included the following sentence in the method sections: “These snack foods are very popular among children in the Netherlands.”

We agree that examining the effects on fruit would be very interesting, but in our opinion the effects of food promotion techniques for fruits are better know than among vegetables. In addition, vegetables are even less consumed than fruits 

C4: The investigators also note that they, “only tested one Instagram post.” It seems unlikely that one post showing a social influencer eating a vegetable once is enough to outweigh the impact of years of inadequate vegetable consumption and advertising of unhealthy food options. Clearly, there are many other possibilities to consider with regard to promotion of healthy diet in children.

A4: We agree completely with the reviewer and therefore have stated that the accumulation of healthy food promotion techniques should be tested. In addition, multiple options are provided in the discussion and conclusion.

Minor Comments

None

Reviewer 2 Report

IJERPH-743981 “The effect of the promotion of vegetables by a social influencer on children’s subsequent vegetable intake: An experimental study”

General comment: The stated aim of the study was to examine, via an experimental design, if promoting vegetables by a popular social influencer on social media 15 (Instagram) increased subsequent actual vegetable intake among children.  The topic has not received much attention in the literature and is yet important in view of wide-spread use of social media by children.  Thus, I applaud the authors’ effort in their attempt to study the topic.  I am however concerned about several key methodological issues and wonder whether they may have prevented the study from making a useful contribution to the literature.

Major specific comments:

  1. The outcome variable of the experiment - It is not clear why one should expect an intervention that promotes red pepper consumption should lead to more positive stronger intent to consume cucumbers and cherry tomatoes, in addition to red peppers (Line 225). More explanation and justification must be added.
  2. What were the theoretical/conceptual bases of parasocial interaction and persuasion knowledge and how were they related to the Social Learning Theory (Line 49), the Processing of Commercial Media Content Model (line 54), the Differential Susceptibility to Media Effects Model (line 74), and/or the Promotion of Healthy Food Model (Line 99)? Clearer discussion must be included. Also, if any of the theories above was/were not related to these measures, then they do not belong to the manuscript.
  3. Line 143 – Embedding the Instagrampost in a paper question questionnaire raises serious suspicion of the validity of the study and its findings. Unlike print materials, Instagram is a dynamic and social communication platform.  Children’s reaction to a post on Instagram therefore can be reasonably expected to differ from that to a post on the paper.  This difference in turn can severely weaken the validity of the study’s assertion that it looked at a promotion effect BY A SOCIAL INFLUENCER.  Therefore, I’d strongly recommend a reconsideration of how the manuscript frame the research topic.
  4. Furthermore, I wonder if the null findings were primarily because any of the three vegetables was not among the commonly consumed vegetables by participants. Though the authors acknowledged this as a limitation of the study, I suspect the problem is more serious than that – the experiment was not given a fair chance to more useful findings because (1) the Instagram post only mentioned red peppers and (2) the three outcome vegetables were not popular with the participants to begin with.  Other vegetables with more frequent consumption should have been used in the study.      

Other specific comments:

  1. Use of the terms “actual” and “intake” – If the outcome variable of the experiment was hypothetical choices, per Line 150, and if no actual consumption was recorded, then the study should not use the term “intake.” An alternative term such as “hypothetical choice” would characterize the outcome more accurately.
  2. Use of the term “vegetables” associated with the intervention – since the Instagram only mentioned red peppers, per Line 145, it is not accurate to characterize the intervention as promoting “vegetables,” which include many products other than red pepper.
  3. Line 152 – should mention the total number of participants here.     

Author Response

General comment: The stated aim of the study was to examine, via an experimental design, if promoting vegetables by a popular social influencer on social media 15 (Instagram) increased subsequent actual vegetable intake among children.  The topic has not received much attention in the literature and is yet important in view of wide-spread use of social media by children.  Thus, I applaud the authors’ effort in their attempt to study the topic.  I am however concerned about several key methodological issues and wonder whether they may have prevented the study from making a useful contribution to the literature.

A: We thank the reviewer for reviewing the manuscript and the positive words. We hope that we have convinced the reviewer that our study adds significantly to the existing literature.

Major specific comments:

C1: The outcome variable of the experiment - It is not clear why one should expect an intervention that promotes red pepper consumption should lead to more positive stronger intent to consume cucumbers and cherry tomatoes, in addition to red peppers (Line 225). More explanation and justification must be added.

A1: In general, food advertising studies have shown that advertisements often lead to spill-over effects. In previous studies that we have conducted we have found the same (see Folkvord et al., 2013, 2014, 2017) , although most of them were for unhealthy foods. Therefore we have decided on beforehand to present different vegetables as well. Next to that, some children do not like red peppers, and would therefore not consume any vegetables as a results. We have now included the following sentences in the methodology section: “We used more vegetables snacks than were shown in the Instagram post because previous studies have shown that food advertising leads to spill-over effects as well [9, 25,26].”

C2: What were the theoretical/conceptual bases of parasocial interaction and persuasion knowledge and how were they related to the Social Learning Theory (Line 49), the Processing of Commercial Media Content Model (line 54), the Differential Susceptibility to Media Effects Model (line 74), and/or the Promotion of Healthy Food Model (Line 99)? Clearer discussion must be included. Also, if any of the theories above was/were not related to these measures, then they do not belong to the manuscript.

A2: We agree with the reviewer that we should better relate the theories that we have used to support the hypotheses with parasocial interaction and persuasion knowledge. We have revised the introduction accordingly.

C3: Line 143 – Embedding the Instagrampost in a paper question questionnaire raises serious suspicion of the validity of the study and its findings. Unlike print materials, Instagram is a dynamic and social communication platform.  Children’s reaction to a post on Instagram therefore can be reasonably expected to differ from that to a post on the paper.  This difference in turn can severely weaken the validity of the study’s assertion that it looked at a promotion effect BY A SOCIAL INFLUENCER.  Therefore, I’d strongly recommend a reconsideration of how the manuscript frame the research topic.

A3: We agree with the reviewer that this raises concerns about the validity. Therefore, we have reframed the study as a pilot-study and provided multiple recommendations that are in line with the comment that the reviewer is making.

C4: Furthermore, I wonder if the null findings were primarily because any of the three vegetables was not among the commonly consumed vegetables by participants. Though the authors acknowledged this as a limitation of the study, I suspect the problem is more serious than that – the experiment was not given a fair chance to more useful findings because (1) the Instagram post only mentioned red peppers and (2) the three outcome vegetables were not popular with the participants to begin with.  Other vegetables with more frequent consumption should have been used in the study.   

A4: We agree with the reviewer that we tested only 1 kind of vegetables in the stimulus materials, which we also have stated as a limitation in the discussion. We also have formulated guidelines for future research. For the second point, the vegetables we have selected in the current study (red peppers, cucumbers, and cherry tomatoes) are the most consumed vegetable snacks in the Netherlands, also among the target population.  We have emphasized on this point in the manuscript.

Other specific comments:

C5: Use of the terms “actual” and “intake” – If the outcome variable of the experiment was hypothetical choices, per Line 150, and if no actual consumption was recorded, then the study should not use the term “intake.” An alternative term such as “hypothetical choice” would characterize the outcome more accurately.

A5: Actual intake was assessed, as we have described in the procedures sections: When the child returned to their seat, the research assistant coded (encrypted) which and how many vegetables were consumed by the child without notifying the children that this was coded”.

C6: Use of the term “vegetables” associated with the intervention – since the Instagram only mentioned red peppers, per Line 145, it is not accurate to characterize the intervention as promoting “vegetables,” which include many products other than red pepper.

A6: We agree with the reviewer that we should be more careful in labeling the intervention with vegetables, but should specify that it is about red peppers. Therefore, we have revised this accordingly in the text.

C7: Line 152 – should mention the total number of participants here.     

A7: We have added this information here.

References:

Folkvord, F., Anschütz, D. J., Buijzen, M., & Valkenburg, P. M. (2013). The effect of playing advergames that promote energy-dense snacks or fruit on actual food intake among children. The American journal of clinical nutrition97(2), 239-245.

Folkvord, F., Anschütz, D. J., Nederkoorn, C., Westerik, H., & Buijzen, M. (2014). Impulsivity,“advergames,” and food intake. Pediatrics133(6), 1007-1012.

Folkvord, F., Anastasiadou, D. T., & Anschütz, D. (2017). Memorizing fruit: The effect of a fruit memory-game on children's fruit intake. Preventive medicine reports5, 106-111.

Reviewer 3 Report

The article entitled “The Effect of the Promotion of Vegetables by a Social Influencer on Children’s Subsequent Vegetable Intake: An Experimental Study” address an interesting topic and is well-written. However, some revisions should made by authors, especially in the methodology section in order to improve the quality of the manuscript. Following, I am going to indicate some revisions or changes that authors should implement in this regard:

Introduction:

In the introduction section, authors should include more information about the prevalence obtained in previous studies regarding fruit and vegetable consumption during childhood. This information should be included at the beginning of this section, in order to establish the need to develop new strategies oriented to the promotion of a healthy diet. Further, a short review about classical interventions oriented to improve diet in this population and their possible limitations should be also addressed in this section.

Furthermore, authors have reviewed previous research employing Instagram or another social network in the promotion of healthy lifestyles in children or other populations. This effectiveness of this previous studies should be explained more deeply in the introduction.

Participants:

Based on the mean age of the participants, it seems that the experiment has been conducted with the participation of adolescents rather than children.

Procedure

Moreover, if it could be possible by copyright, it could be helpful for future readers, having the pictures of the employed posts in the research. It could be important the format in which the post was developed and presented. A big difference between the presented posts in paper and the original ones included in Instagram in other format (paper vs audiovisual format) could entails a limitation. Authors should had considered the exposure to an original post in Instagram.

How much time were children exposed to the posts? I assume that the time that they had to eat any product in any condition was the same, but this fact should be included and explained. Children eat individually or in a group? Could be children influenced by the food choice of other participants in the group?

Probably, a methodology in which children were exposed individually to an audiovisual post and then, have dishes with the same proportion of types of food (healthy and snacks) fit better with the aim of the study.

General diet preferences of evaluated children were identified? The diet that they usually follow were analyzed? The inclusion of a Food Frequency Questionnaire adapted for children would be interesting in this regard.

All ad-hoc and specific developed items employed to evaluate any variable should be included in the measures section.

Results:

Table of Pearson correlations should be included in the manuscript.

Discussion:

The discussion section is adequate and justify possible underlying mechanisms to explain the obtained results.

Author Response

C1: The article entitled “The Effect of the Promotion of Vegetables by a Social Influencer on Children’s Subsequent Vegetable Intake: An Experimental Study” address an interesting topic and is well-written. However, some revisions should made by authors, especially in the methodology section in order to improve the quality of the manuscript. Following, I am going to indicate some revisions or changes that authors should implement in this regard:

A1: We thank the reviewer for reviewing our manuscript and the positive words. In addition, we will revise the manuscript according the suggestions the reviewer has made to improve the quality of the paper.

Introduction:

C2: In the introduction section, authors should include more information about the prevalence obtained in previous studies regarding fruit and vegetable consumption during childhood. This information should be included at the beginning of this section, in order to establish the need to develop new strategies oriented to the promotion of a healthy diet. Further, a short review about classical interventions oriented to improve diet in this population and their possible limitations should be also addressed in this section.

A2: We now have included a section at the beginning of the introduction describing this information.

C3: Furthermore, authors have reviewed previous research employing Instagram or another social network in the promotion of healthy lifestyles in children or other populations. This effectiveness of this previous studies should be explained more deeply in the introduction.

A3: We now have elaborated more on these studies.

Participants:

C4: Based on the mean age of the participants, it seems that the experiment has been conducted with the participation of adolescents rather than children.

A4: We agree with the reviewer, therefore we have revised it throughout the manuscript.

Procedure

C5: Moreover, if it could be possible by copyright, it could be helpful for future readers, having the pictures of the employed posts in the research. It could be important the format in which the post was developed and presented. A big difference between the presented posts in paper and the original ones included in Instagram in other format (paper vs audiovisual format) could entails a limitation. Authors should had considered the exposure to an original post in Instagram.

A5: We agree with the reviewer that it would be useful to publish the Instagram post, but because we have used an existing post and do not have the authorship to publish them ourselves, we chose to not publish the images. Of course, we understand that presenting posts in paper instead of original ones online is a limitation, therefore we have included this as a limitation in the discussion. We have described it in the manuscript as follows: “A second limitation is that we have used a pepr format for the Instagram posts, while in reality adolescents watch Instagram posts online with an audiovisual format. Therefore, recommendations for future research is that adolescents should be exposed to multiple Instagram posts, in an audiovisual format, thereby taking into account the similarity of the content of the stimulus materials for the different conditions.”

C6: How much time were children exposed to the posts? I assume that the time that they had to eat any product in any condition was the same, but this fact should be included and explained. Children eat individually or in a group? Could be children influenced by the food choice of other participants in the group? Probably, a methodology in which children were exposed individually to an audiovisual post and then, have dishes with the same proportion of types of food (healthy and snacks) fit better with the aim of the study.

A6: Adolescents were exposed to the posts for 1-2 minutes, on average, which is now included in the description in the methodology. This is comparable to normal exposure time on Instagram itself. The eating was in class, and this could of course influence their food choice. Nonetheless, this is also something that occurs in real-life, thereby increasing the ecological validity of the study. We have stated in the discussion the following: “Future studies should examine the effects of an audiovisual Instagram post promoting vegetables on the intake of an individual child or adolescent, without other children around that could potentially influence their consumption behavior.”

C7: General diet preferences of evaluated children were identified? The diet that they usually follow were analyzed? The inclusion of a Food Frequency Questionnaire adapted for children would be interesting in this regard.

A7: General diet preferences or usual diets were not analyzed. Including the Food Frequency Questionnaire would be interesting indeed, but we did not take this into account. The full experiment took around 20 minutes per class, during school hours, so unfortunately we needed to decide which questions we wanted to present and which not. Nonetheless, we included the Food Frequency Questionnaire for children as a recommendation for future studies in this area.

C8: All ad-hoc and specific developed items employed to evaluate any variable should be included in the measures section.

A8: We have included this information in the manuscript.

Results:

C9: Table of Pearson correlations should be included in the manuscript.

A9: Considering that we found no significant correlations between the variables, common practice is to not include the table with Pearsons correlations because it does not inform the reader. If the reviewer argues that it is still important to include the table of Pearsons correlations, and the editor agrees, we will of course integrate the table.

Discussion:

C10: The discussion section is adequate and justify possible underlying mechanisms to explain the obtained results.

A10: We thank the reviewer for this positive comment.

Round 2

Reviewer 2 Report

Thanks to the authors for considering my comments. I consider the revision is satisfactory and responsive.  I also appreciate the effort to include actual comments to illustrate parasocial interaction and persuasion knowledge.  Below are a few more minor suggestions for your consideration.

Line 15 - I think you can call it an "experimental pilot study" or something like that to note the nature of the study.

line 175 - it may be useful to elaborate a little what spill-over effects mean in the context of this study.

lines 289-294 - the description of the study aim is inconsistent with that at lines 149-153.

Author Response

C1: Thanks to the authors for considering my comments. I consider the revision is satisfactory and responsive.  I also appreciate the effort to include actual comments to illustrate parasocial interaction and persuasion knowledge.  Below are a few more minor suggestions for your consideration.

Line 15 - I think you can call it an "experimental pilot study" or something like that to note the nature of the study.

line 175 - it may be useful to elaborate a little what spill-over effects mean in the context of this study.

lines 289-294 - the description of the study aim is inconsistent with that at lines 149-153

A1: We thank the reviewer for reviewing the manuscript again, and of course for the positive words. In addition, we have revised the minor textual suggestions that the reviewer suggested. 

Reviewer 3 Report

The authors have adressed the proposed commentaries and changes in the manuscript. Just to consider the inclusion of the table of correlations between variables and, moreover, the inclusion of indicators of size effect in the analysis of differences. This fact is necessary to the study be included in future systematic reviews and meta-analysis. 

Author Response

C1: The authors have adressed the proposed commentaries and changes in the manuscript. Just to consider the inclusion of the table of correlations between variables and, moreover, the inclusion of indicators of size effect in the analysis of differences. This fact is necessary to the study be included in future systematic reviews and meta-analysis. 

A1: We thank the reviewer again for reviewing the manuscript, and for the positive words. In addition, we have included the effect sizes for the analyses of differences, considering the future meta-analyses.